# EraseLoRA: MLLM-Driven Foreground Exclusion and Background Subtype Aggregation for Dataset-Free Object Removal

## Abstract

Object removal requires more than erasing a target—it must reconstruct the missing region with high structural fidelity while preserving diverse background context. Existing diffusion-based dataset-free approaches attempt to redirect self-attention away from the masked target but fail in two critical ways: (1) non-target foregrounds are often misinterpreted as background, causing unintended object regeneration, and (2) disruption of short-range activations degrades fine details and prevents coherent integration of multiple background cues. We introduce **EraseLoRA**, a dataset-free object-removal framework that leverages the visual reasoning power of **multimodal large-language models (MLLMs)** to exclude foreground distractions and assemble rich background content. The first stage, **BRF (Background Reconstruction with Foreground Exclusion)**, isolates and removes non-target objects through MLLM-guided reasoning on a single image–mask pair, producing clean background candidates without ground-truth supervision. The second stage, **Background Subtype Aggregation (BSA)**, restores the masked region by treating each inferred background subtype as a puzzle piece, enforcing their consistent integration to preserve both local detail and global context. EraseLoRA achieves state-of-the-art object-removal performance across diverse diffusion backbones **without any additional training data or ground-truth background**, demonstrating that MLLM reasoning—applied here for structural reconstruction rather than object generation—can directly guide diffusion models to rebuild complex scenes from a single image with unprecedented structural and contextual coherence.

## 1 Introduction

With the development of diverse generative models, the field of image generation has advanced significantly to enable the synthesis of realistic and high-quality images. Early approaches leveraged generative adversarial networks (GANs) (Goodfellow et al., 2021), which enable fast single-step image generation but often suffer from instability, blurring, and visual artifacts. Recently, diffusion models (DMs) have emerged as a powerful alternative of GANs, generating more consistent and high-fidelity images through a multi-step denoising process (Ho et al., 2020; Rombach et al., 2022). In particular, Text-to-Image (T2I) diffusion models like Stable Diffusion (Rombach et al., 2022) enable more controllable synthesis through conditions such as text prompts and reference images, and have become the dominant backbone for various image generation tasks (Rombach et al., 2022; Podell et al., 2023; Esser et al., 2024). Image inpainting, a key task of this field, aims to reconstructs missing region based on the surrounding visible context. Prior works (Avrahami et al., 2021) have finetuned T2I diffusion models (Rombach et al., 2022; Xie et al., 2022; Zhuang et al., 2024) with text condition, so that the masked region can be filled according to the wanted intent.

Object removal is a representative subtype of inpainting whose goal is to remove the masked target object and to reconstruct background without the regeneration of object-like artifacts. However, previous finetuned T2I diffusion models (Rombach et al., 2022; Xie et al., 2022) for inpainting are unsuitable for object removal as they primarily aim to generate plausible new object inside the mask. Therefore, effective object removal requires either additional training signals or strategies. Existing object removal methods can be grouped into two categories: Dataset-Driven and Dataset-

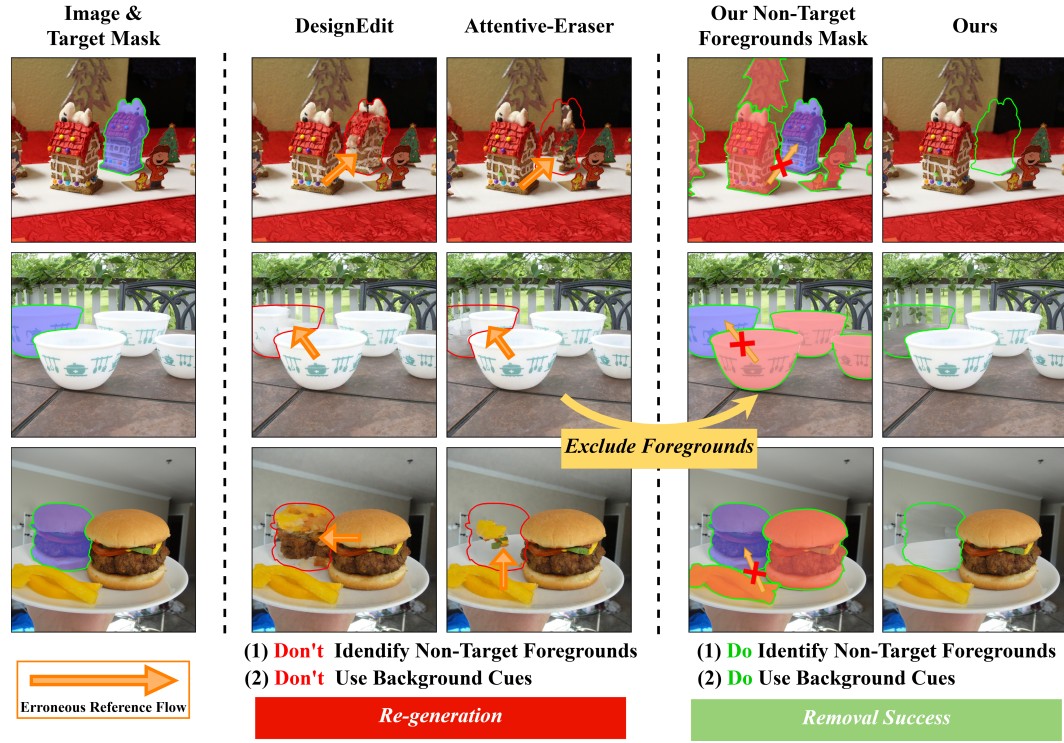

Figure 1: **Qualitative comparison** between existing dataset-free SOTA methods (DesignEdit, Attentive-Eraser) and our proposed **EraseLoRA** framework. Previous methods often misinterpret non-target foregrounds as background, leading to regeneration, as they simply reference unmasked regions without explicit background cues. In contrast, our method explicitly identifies and excludes non-target foregrounds while leveraging background cues, resulting in accurate background filling and clean object removal.

Free. Dataset-driven methods (Zhuang et al., 2024; Liu et al., 2025; Jiang et al., 2025) generally finetune diffusion models with additional paired training dataset—images before and after object removal. Despite their effectiveness, these methods require paired dataset difficult to construct at scale. Therefore, dataset-free methods (Ekin et al., 2024; Jia et al., 2025; Sun et al., 2025) based on powerful pretrained T2I diffusion models have recently been proposed to overcome these limitations. Recent state-of-the-art (SOTA) approaches commonly redirect self-attention by suppressing masked regions and enhancing unmasked regions for object removal, a strategy we term hard self-attention suppression (HSAS), thereby encouraging the model to focus on surrounding context rather than masked context.

While the HSAS methods alleviate the generation of a new object, they still have two critical limitations. First, they are background-unaware: by treating all unmasked pixels as background to reference, they often misinterpret non-target foreground objects as background, causing unintended object regeneration as shown in Fig. 1. To prevent this, identifying pure background without any foreground objects is crucial to achieve successful object removal. Second, HSAS methods disrupt short-range activations of self-attention, referring to the local interactions where latent tokens mainly attend to their nearby neighbors. These activations are crucial for preserving fine-grained details, and their disruption leads to blur artifacts. Furthermore, recent diffusion transformers DiT T2I diffusion models (Esser et al., 2024) (e.g., FLUX, SD3.5) adopt a patchified latent representation that enables efficient computation and has become a widely used backbone with strong generative performance. However, when HSAS is applied to such architectures, it further yields checkerboard-like group-wise artifacts due to instability in patch-wise attention computation (Fig. 2). These structural artifacts show that SOTA methods are neither detail-preserving nor robust when extended to recent powerful DiT T2I diffusion architectures.

We propose a dataset-free object removal framework EraseLoRA, which explicitly excludes all foreground distractions by utilizing multimodal large-language models (MLLMs) and aggregates diverse background context in two stages: (1) Background Reconstruction with Foreground Ex-

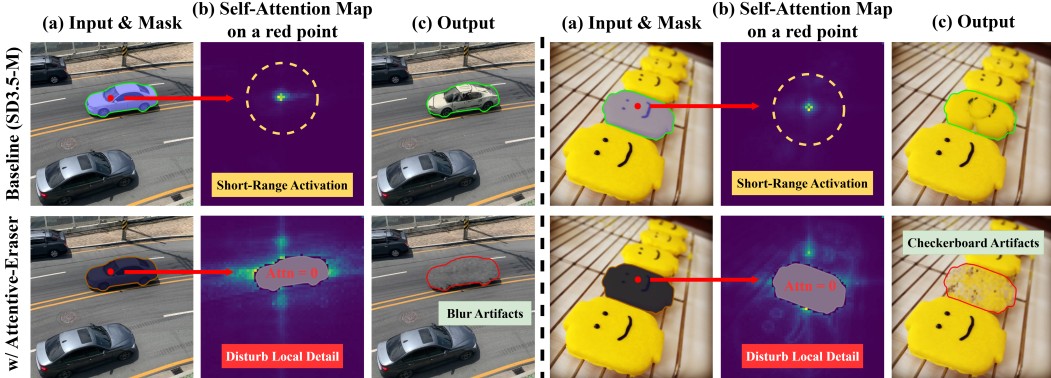

Figure 2: Limitations of SOTA methods. Comparison between the SD3.5-M baseline (top) and the same backbone with Attentive-Eraser (bottom). While the baseline preserves short-range self-attention within the mask, enabling detailed and stable reconstruction, Attentive-Eraser suppresses local self-attention. This disruption breaks fine details and causes blur artifacts (left) as well as checkerboard-like group-wise artifacts on DiT T2I diffusion backbone (right).

clusion(BRF):, we isolate non-target objects through MLLM-guided visual reasoning on a single image–mask pair, predicting clean background candidates without ground-truth supervision. (2) Background Subtype Aggregation (BSA): each predicted background subtype is treated as a puzzle piece for filling the masked region, and their consistent integration ensures accurate background completion while preserving both local detail and global context. Through foregrounds exclusion and background subtype aggregation, EraseLoRA outperforms previous SOTA dataset-free methods in both quantitative and qualitative evaluations across diverse T2I diffusion backbones.

The main contributions of our method can be summarized as follows:

- We first observe that the visual reasoning power of MLLMs in distinguishing foreground and background cues consistently improves object removal performance, even when integrated into existing state-of-the-art approaches.

- We propose EraseLoRA, dataset-free object-removal framework adopts the concept of test-time adaptation, guide diffusion model to effectively assemble background cues with structural and contextual coherence without any ground-truth label through a newly introduced Cross-Attention Puzzle Loss.

- EraseLoRA outperforms prior state-of-the-art methods across multiple metrics, achieving superior quality and robustness.

## 2 RELATED WORK

### 2.1 IMAGE INPAINTING WITH GENERATIVE MODELS

Image Inpainting, one of the main task of image generation field, aims to reconstruct missing regions consistently based on visible context. Early GAN-based approaches (Zhao et al., 2021; Zuo et al., 2023; Sargsyan et al., 2023) trained a generator to fool a discriminator, but their single-step generation often led to instability, blurring, and artifacts. In response to these challenges, diffusion models (DMs) (Xie et al., 2022; Yang et al., 2022) have emerged as a powerful alternative, generating more detailed and high-fidelity images through a iterative denoising process. Text-to-image (T2I) diffusion models (Manukyan et al., 2023; Xie et al., 2022) such as Stable Diffusion further enable controllable synthesis via external conditions like text prompts and reference images.

Building on this controllability, recent works have further incorporated the reasoning capabilities of MLLMs into T2I diffusion models, allowing more fine-grained control over the content generated within masked regions (Fanelli et al., 2025; Tianyidan et al., 2025; Zhou et al., 2025). For example, MLLMs have been employed to optimize user's instructions, or propose descriptions that align with the visible context. However, as illustrated in Fig. 3, these applications of MLLMs have been limited to new content generation or holistic image understanding. We first observe and leverage the visual

Table 1: Conceptual Comparison of our method with recent approaches.

| Properties | Image Inpainting | | Image Inpainting for Object Removal | | | | | | | |
| --- | --- | --- | --- | --- | --- | --- | --- | --- | --- | --- |
| | [WACV'25] I Dream My Painting | [AAAI'25] Anywhere | [ECCV'24] PowerPaint | [NeurIPS'25] CLIPAway | [CVPR'25] Entity-Erasure | [CVPR'25] Erase-Diffusion | [CVPR'25] Smart-Eraser | [AAAI'25] Design-Edit | [AAAI'25] Attentive-Eraser | **Ours** |
| **Dataset-Free** | ✗ | ✓ | ✗ | ✓ | ✗ | ✗ | ✗ | ✓ | ✓ | ✓ |
| Identifies **non-target foregrounds** across classes | ✗ | ✗ | ✗ | ✗ | ✗ | ✗ | ✗ | ✗ | ✗ | ✓ |
| Uses MLLM for content generation | ✓ | ✓ | ✗ | ✗ | ✗ | ✗ | ✗ | ✗ | ✗ | ✓ |
| Uses MLLM for **fill-in background prediction** | ✗ | ✗ | ✗ | ✗ | ✗ | ✗ | ✗ | ✗ | ✗ | ✓ |
| Enforces consistent **background aggregation** | ✗ | ✗ | ✗ | ✗ | ✗ | ✗ | ✗ | ✗ | ✗ | ✓ |
| Model-agnostic under attention refinement | ✓ | ✗ | ✗ | ✗ | ✗ | ✗ | ✗ | ✗ | ✗ | ✓ |

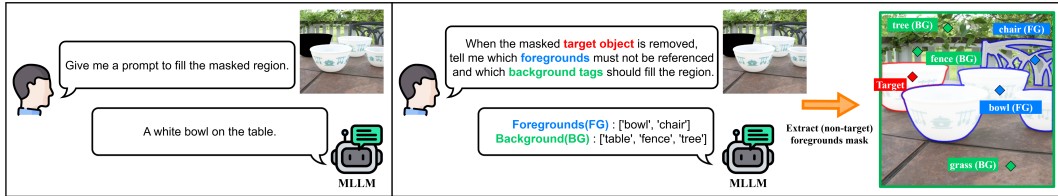

Figure 3: Comparison of MLLM usage between previous works and our EraseLoRA. (Left) Prior works primarily employ MLLMs as content generators, predicting plausible objects within the mask for object generation. (Right) In contrast, EraserLoRA leverages the visual reasoning power of MLLMs to exclude non-target foregrounds and infer background cues for reconstructing the masked region, which are further used to separate foreground and background areas.

reasoning capability of MLLMs to distinguish foreground distractions and background cues which can be used as restoration after target elimination.

## 2.2 OBJECT REMOVAL DIFFUSION MODELS

Object removal, a representative subtype of inpainting, goes beyond simply erasing a masked target and must reconstruct the masked region with structural fidelity and coherent background context. Existing object removal methods can be broadly grouped into Dataset-Driven and Dataset-Free. Dataset-driven approaches primarily adopt supervised learning, training models with paired images before and after object removal. *PowerPaint* (Zhuang et al., 2024) leverages learnable task prompts using positive and negative controls for object removal. *SmartEraser* (Jiang et al., 2025) integrates a visual embedding of the masked object with a predefined removal prompt as guidance. *EraseDiffusion* (Liu et al., 2025) redefines and trains the object removal pathway from a noisy input to post-removal image. Such pairs are typically constructed either by synthesizing target objects into images or by extracting them from video frames. However, synthesized images are often unrealistic and video-based collection is costly, making large-scale dataset construction challenging.

To overcome these limitations, dataset-free approaches leverage pretrained T2I diffusion models without additional supervision. *CLIPAway* (Ekin et al., 2024) constructs a background-focused embedding to suppress foreground information and uses as guidance. More generally, most methods rely on hard self-attention suppression (HSAS), suppressing attention to masked tokens while redirecting it to visible context. *DesignEdit* (Jia et al., 2025) adopts a key-masking strategy that suppresses all keys within the masked region, forcing the model to focus only on the unmasked region. *AttentiveEraser* (Sun et al., 2025) leverages the difference between the noise predicted by HSAS and the original model's predicted noise. This core mechanism has become a widely adopted strategy, not only in dataset-free settings but also in many dataset-driven approaches to ensure effective removal. For example, some methods introduce self-attention suppression in specific regions guided by learned priors such as text embeddings or segmentation priors (Li et al., 2024; Zhu et al., 2025).

Although HSAS methods alleviate unwanted object generation, they remain background-unaware, often misinterpreting non-target foregrounds as background, and they disrupt short-range self-attention activation, leading to detail loss and poor robustness across diverse diffusion backbones. In contrast, we leverage the visual reasoning ability of MLLMs to explicitly exclude non-target fore-

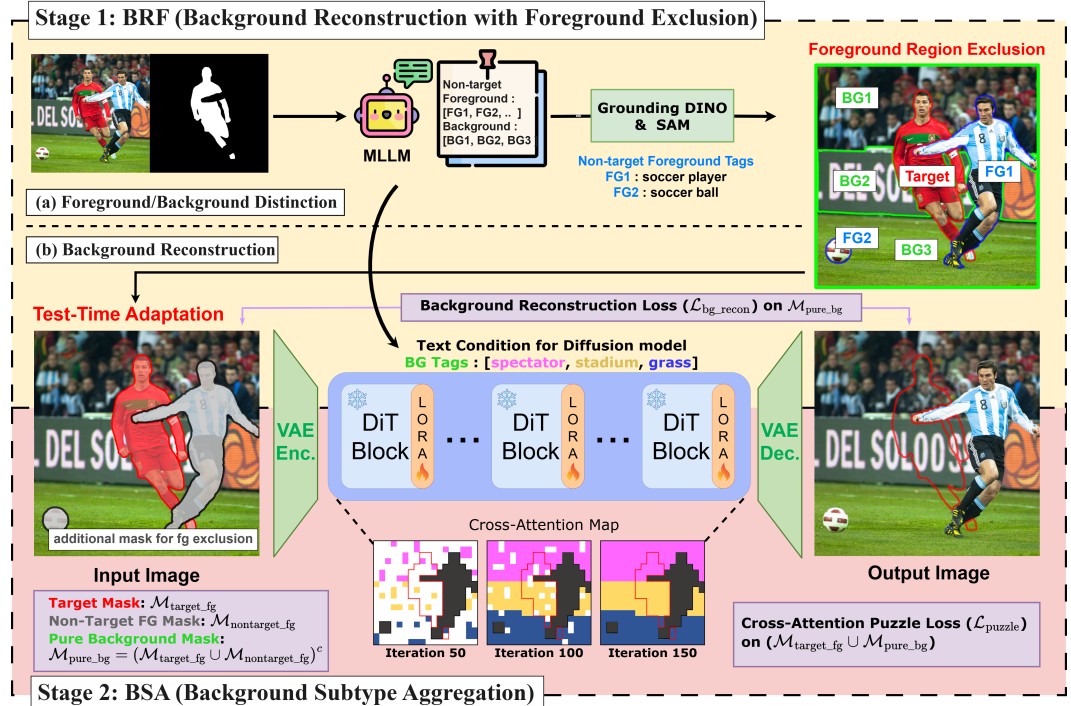

Figure 4: **Overview of our EraseLoRA.** Stage 1: **BRF** excludes non-target foregrounds and reconstructs pure background regions via a Background Reconstruction Loss. Stage 2: **BSA** integrates MLLM-predicted background subtypes into the mask using a Cross-Attention Puzzle Loss, achieving coherent reconstruction of local details and global structure. Both losses are applied simultaneously during test-time adaptation, guiding the diffusion model toward coherent background completion.

grounds and coherently aggregate background cues, enabling structurally consistent object removal under cross-attention refinement.

## 3 METHOD

Our proposed EraseLoRA, a dataset-free object removal framework, consists of two stages: (1) Background Reconstruction with Foreground Exclusion (BRF), and (2) Background Subtype Aggregation (BSA). The general framework diagram of EraseLoRA is illustrated in Fig.4 and each stage will be described in detail below.

### 3.1 BACKGROUND RECONSTRUCTION WITH FOREGROUND EXCLUSION(BRF)

The first stage, BRF, prevents unintended regeneration of objects by explicitly excluding non-target foregrounds and reconstructing pure background regions. It consists of two substages:

**(a) Foreground/Background Distinction.** Given an input image–mask pair, we leverage the visual reasoning power of MLLMs to generate semantic tags distinguishing non-target foregrounds from background. For each foreground tag, detection and segmentation modules (Grounding DINO (Liu et al., 2024) and SAM (Ravi et al., 2025)) are used to localize its region. The union of these localized regions defines the non-target foreground mask, while the pixels outside both target and non-target foregrounds define the pure background region. The identified background cues—both regions and semantic tags—serve as reliable guidance, with regions providing supervision for reconstruction and tags guiding the subsequent filling process.

**(b) Background Reconstruction.** Existing HSAS methods disrupt short-range self-attention, causing local detail loss and structural artifacts. To overcome this, we introduce a test-time adaptation (TTA) scheme tailored for object removal. In general, TTA adapts a pretrained model to each test

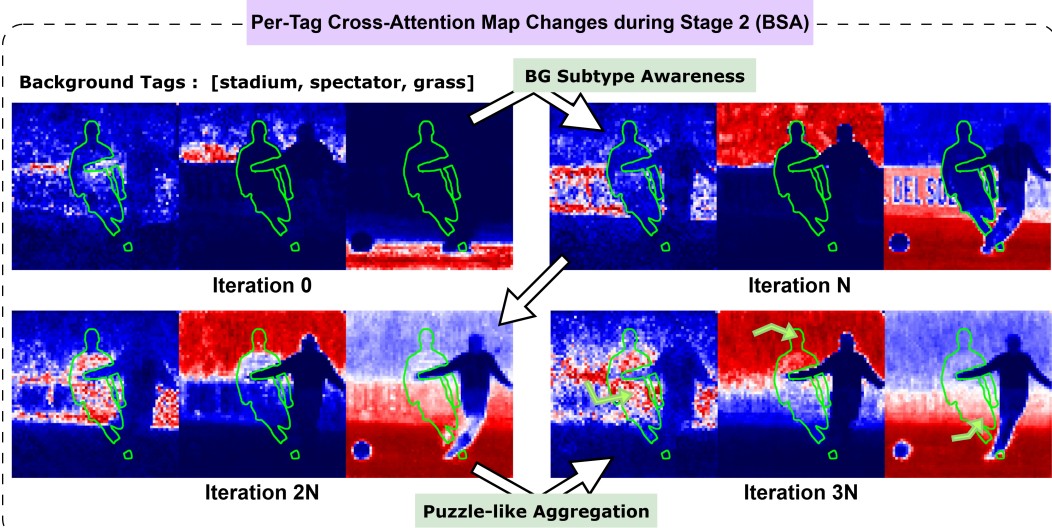

Figure 5: **Per-tag cross-attention map changes** in Stage 2 (BSA). With background tags (e.g., "stadium, spectator, grass"), the model recognizes background subtypes (top) and fills the masked region in a puzzle-like manner (bottom), achieving fine-grained semantic alignment and coherent background completion.

input during inference using self-supervised objectives (e.g., in classification or segmentation), but such supervision is unavailable inside the mask for removal task. We address this by exploiting the pure background regions identified in Stage 1(a) as pseudo ground-truth, and define a Background Reconstruction Loss ($\mathcal{L}_{\text{bg\_recon}}$). Unlike HSAS methods, this loss softly guides self-attention at the noise level (see Appendix A for details on diffusion and attention mechanisms), preserving activations rather than suppressing them, which maintains fine local details and ensures robustness across diverse T2I diffusion backbones.

Formally, $\mathcal{L}_{\text{bg\_recon}}$ encourages faithful recovery of the input background within the pure background region $M_{\text{pure\_bg}}$:

$$\mathcal{L}_{\text{bg\_recon}} = \frac{1}{|M_{\text{pure\_bg}}|} \sum_{p \in M_{\text{pure\_bg}}} \|\hat{I}[p] - I[p]\|_2^2, \tag{1}$$

where $p$ indexes pixels, $I$ denotes the original input image, and $\hat{I}$ denotes the reconstructed output.

## 3.2 Background Subtype Aggregation (BSA)

Although BRF enables supervision in pure background regions via the Background Reconstruction Loss, the masked region itself still lacks ground-truth labels. In practice, simply injecting background tags as text prompts does not guarantee that the masked area will be faithfully filled with those subtypes; instead, random objects or textures inconsistent with the scene are often generated.

To overcome this limitation, BSA explicitly controls how the masked area is reconstructed by enforcing that it is composed only of valid background subtypes predicted by the MLLM. In this way, cross-attention is guided not just toward local fidelity but also toward global semantic structure, ensuring that the completed background is both fine-grained and contextually coherent. We introduce a Cross-Attention Puzzle Loss, which treats each background subtype as a "puzzle piece" and guides their integration within the masked region (Fig. 5). The Cross-Attention Puzzle Loss combines the alignment and diversity terms as $\mathcal{L}_{\text{puzzle}} = \mathcal{L}_{\text{align}} + \mathcal{L}_{\text{div}}$.

**(1) Background Alignment Loss.** Let $A_t^{\text{cross}} \in \mathbb{R}^{H_{\text{cross}} \times W_{\text{cross}}}$ denote the cross-attention map corresponding to a background tag $t \in \mathcal{T}_{\text{bg}}$. We sum across tags to obtain a reconstructed attention map $A^{\text{recon}}$, and enforce that it is concentrated only within the union of pure background and target mask:

$$A^{\text{recon}}[p] = \sum_{t \in \mathcal{T}_{\text{bg}}} A_t^{\text{cross}}[p], \quad \mathcal{L}_{\text{align}} = 1 - \text{Dice}(A^{\text{recon}}, M_{\text{pure\_bg}} \cup M_{\text{target\_fg}}). \tag{2}$$

**(2) Tag Diversity Loss.** To avoid collapse into a single dominant subtype, we require that every background tag contributes within the target mask $M_{\text{target\_fg}}$. For each tag $t$, let $S_t$ denote the total activation inside the target region:

$$|S_t| = \sum_{p \in M_{\text{target\_fg}}} A_t^{\text{cross}}[p], \quad \mathcal{L}_{\text{div}} = 1 - \min_{t \in \mathcal{T}_{\text{bg}}} |S_t|. \tag{3}$$

Finally, EraseLoRA integrates the Background Reconstruction Loss from BRF with the Cross-Attention Puzzle Loss from BSA, forming the overall TTA loss as $\mathcal{L}_{\text{TTA}} = \mathcal{L}_{\text{bg\_recon}} + \lambda \mathcal{L}_{\text{puzzle}}$. $\lambda$ controls the contribution of the cross-attention puzzle loss term. This joint objective ensures both local background fidelity (via $\mathcal{L}_{\text{bg\_recon}}$) and subtype-consistent reconstruction (via $\mathcal{L}_{\text{puzzle}}$).

## 4 EXPERIMENTS

### 4.1 IMPLEMENTATION DETAILS

We evaluate EraseLoRA on both UNet (SDXL (Podell et al., 2023)) and DiT (SD3.5-M (Esser et al., 2024)) T2I diffusion backbones to verify its generalization across various diffusion model architectures. All backbone weights are frozen; only lightweight LoRA adapters in attention blocks are optimized during test-time adaptation (TTA). For each test sample, we run TTA for a small number of iterations using the Final Test-Time Adaptation Loss in Section 4.2. Detailed optimizer settings and implementation details are provided in Appendix B.

### 4.2 EXPERIMENTAL SETUP

**Baseline.** We compare our method to a variety of previous approaches, including both dataset-driven methods (*SDXL-Inpainting* (Podell et al., 2023), *FLUX.1-Fill-dev* (Labs, 2023), *OmniEraser* (Wei et al., 2025), *SmartEraser* (Jiang et al., 2025), *EntityErasure* (Zhu et al., 2025)) and dataset-free methods (*Stable Diffusion 3.5-M* (Esser et al., 2024) +*DesignEdit* (Jia et al., 2025),+*AttentiveEraser* (Sun et al., 2025)). All baselines are reproduced using official implementations when available, or re-implemented according to the descriptions in their papers.

**Test Datasets.** We evaluate our method on two benchmark datasets: OpenImages V7 and ROAD. From OpenImages V7, we randomly sample 1,000 images with object segmentation masks. Since OpenImages V7 provides object annotations but does not contain corresponding object-removed ground-truth images, we treat it as an *unpaired* dataset for evaluation. In contrast, the ROAD dataset consists of 343 videos with paired before/after object-removal frames, from which we curate paired samples suitable for pixel-level evaluation.

**Evaluation Metrics.** For OpenImages V7 (unpaired), pixel-level fidelity cannot be computed. Instead, we use semantic and perceptual metrics from pretrained models, such as DINO (Caron et al., 2021) and CLIP (Radford et al., 2021). We denote by "F" the masked target region, and by "B" the pure background region, i.e., the unmasked area after excluding non-target foregrounds identified by our method. (1) F-DINO / F-CLIP measure similarity between the reconstructed masked region and target object embeddings (lower is better). (2) B-DINO / B-CLIP measure similarity between the reconstructed masked region and pure background embeddings (higher is better). For ROAD (paired), we report pixel-level reconstruction metrics: (1) SSIM and PSNR for structural similarity and signal-to-noise ratio; (2) LPIPS for perceptual distance. For SSIM/PSNR higher is better, while for LPIPS lower is better.

### 4.3 QUANTATIVE AND QUALITATIVE ANALYSIS

The quantitative analysis results are shown in Tab. 2. First, F-DINO and F-CLIP show that most methods can suppress the generation of new object within the mask, but these scores do not reveal subtle artifacts such as unintended references to non-target foregrounds, meaning they cannot fully indicate whether removal is complete. Second, B-DINO and B-CLIP highlight the strength of EraseLoRA, as it achieves much better alignment with pure background regions, confirming the benefit of explicitly excluding non-target foregrounds using MLLMs and TTA Loss. Third, while SSIM and PSNR indicate similar structural fidelity across methods, EraseLoRA achieves the lowest LPIPS, meaning it produces more natural-looking results with fewer visual artifacts. Finally,

Table 2: Quantitative comparison with other methods on test datasets. The best results are highlighted in **bold**, and the second-best results are underlined.

| Method | Param. | OpenImages V7 | | ROAD | | |
|---|---|---|---|---|---|---|
| | | F-DINO↓ / B-DINO↑ | F-CLIP↓ / B-CLIP↑ | SSIM↑ | PSNR↑ | LPIPS↓ |
| *Dataset-Free Approaches:* | | | | | | |
| Stable Diffusion 3.5-M | 2,243 M | | | | | |
| + DesignEdit (AAAI'25) | 2,243 M | 0.440/0.469 | 0.562/0.676 | **0.701** | **18.893** | 0.251 |
| + AttentiveEraser (AAAI'25) | 2,243 M | 0.439/0.457 | **0.559**/0.663 | 0.700 | 18.831 | 0.258 |
| **+ Ours** | **2,255 M** | 0.438/**0.598** | 0.564/**0.749** | 0.699 | 18.734 | **0.235** |
| *Dataset-Driven Approaches:* | | | | | | |
| SDXL-Inpainting | 2,568 M | 0.510/0.446 | 0.623/0.637 | 0.532 | 15.332 | 0.551 |
| FLUX.1-Fill-dev | 11,902 M | 0.806/0.428 | 0.803/0.586 | 0.694 | 16.74 | 0.270 |
| OmniEraser | 16,961 M | **0.425**/0.527 | 0.565/0.705 | 0.554 | 18.237 | 0.309 |
| SmartEraser (CVPR'25) | 1,494 M | 0.563/0.530 | 0.634/0.659 | 0.577 | 18.810 | 0.269 |
| EntityErasure (CVPR'25) | 2,607 M | 0.505/0.562 | 0.582/0.680 | 0.589 | 18.183 | 0.292 |

since F-DINO and F-CLIP alone cannot fully capture artifact leakage, we include qualitative results (Fig. 6 and Appendix D). These show that competing methods often regenerate fragments of removed objects or introduce unrealistic textures, whereas EraseLoRA achieves cleaner and more contextually consistent background restoration.

## 4.4 ABLATION STUDY

To validate the effectiveness and contribution of the proposed EraseLoRA, we conduct ablation studies for analyzing the contributions of each component in Tab. 3, Tab. 4.

Table 3: Ablation study of different architectures and loss components

| Architecture | Method | Loss Components | | Metrics | | Computational Cost | |
|---|---|---|---|---|---|---|---|
| | | $L_{BRF}$ | $L_{BSA}$ | F-DINO/B-DINO | F-CLIP/B-CLIP | VRAM | Param |
| Stable Diffusion 3.5-M | (a) | ✗ | ✗ | 0.900/0.414 | 0.895/0.584 | 21.9 GB | 2,243M |
| | (b) | ✓ | ✗ | 0.510/0.725 | 0.594/0.847 | 52.3 GB | +11.9M |
| | (c) | ✗ | ✓ | 0.421/0.508 | 0.528/0.732 | 52.3 GB | +11.9M |
| | (d) | ✓ | ✓ | 0.448/0.722 | 0.546/0.848 | 52.3 GB | +11.9M |
| Stable Diffusion XL | (a) | ✗ | ✗ | 0.847/0.404 | 0.878/0.532 | 12.4 GB | 2,568M |
| | (b) | ✓ | ✗ | 0.477/0.508 | 0.560/0.178 | 23 GB | +23.2M |
| | (c) | ✗ | ✓ | 0.532/0.474 | 0.587/0.693 | 23 GB | +23.2M |
| | (d) | ✓ | ✓ | 0.462/0.537 | 0.563/0.734 | 23 GB | +23.2M |

**Contribution of TTA.** When neither BRF nor BSA is applied, baselines tend to regenerate objects within the masked region. Applying only the BRF Loss mitigates this issue but still often results in unintended object regeneration. Using BSA alone effectively suppresses regeneration, but introduces blur artifacts and loses fine local details. Combining both achieves the best results, jointly suppressing regeneration and preserving details. Furthermore, as shown in Tab. 3, our method remains stable across different T2I diffusion backbones (SDXL and SD3.5-M), demonstrating the model-agnostic robustness of our TTA.

**Contribution of MLLM.** Tab. 4 demonstrates that incorporating MLLM-derived foreground masks into HSAS methods effectively suppresses unintended regeneration and enhances overall performance. This confirms that explicit exclusion of non-target foregrounds serves as a strong complement to existing dataset-free approaches.

Table 4: Effect of MLLM-guided non-target foregrounds exclusion in dataset-free methods

| Model | w/o FG mask | | w/ FG mask | |
|---|---|---|---|---|
| | F-DINO | B-DINO | F-DINO | B-DINO |
| SDXL | 0.811 | 0.463 | 0.854 (-5.3%) | 0.450 (-2.8%) |
| DesignEdit | 0.682 | 0.531 | 0.521 (+23.6%) | 0.586 (+10.4%) |
| AttentiveEraser | 0.596 | 0.636 | 0.488 (+18.1%) | 0.697 (+9.6%) |

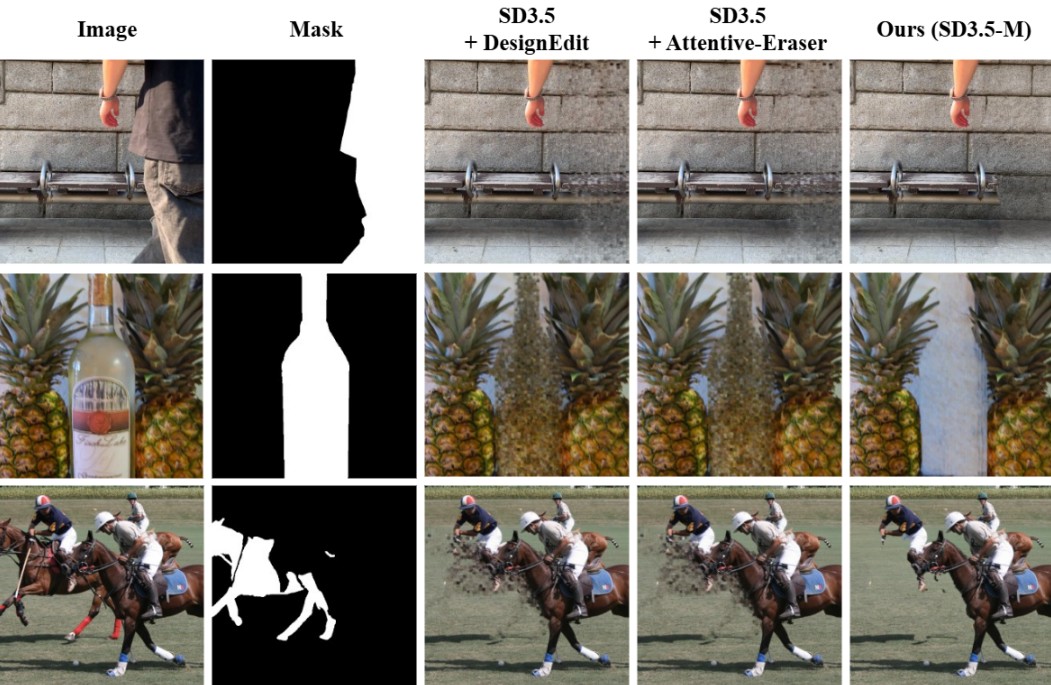

Figure 6: **Comparison with previous SOTA methods**

### 4.5 DISCUSSION

**Insight.** Our study highlights two contributions: first using MLLMs as explicit background separators for object removal, and employing TTA to align diffusion with background cues. Together, these show that semantic reasoning and adaptive optimization can substantially enhance dataset-free object removal.

**Limitations and Future Work.** MLLM-guided reasoning enables explicit non-target foreground exclusion, but remains dependent on the reliability of external priors and may fail in ambiguous cases (e.g., the same person labeled as both background and foreground). Meanwhile, TTA improves removal quality but requires high VRAM and is not entirely training-free. Future work includes developing approaches that can separate foreground and background without relying on MLLMs, exploring lightweight latent optimization to reduce memory cost, and extending the framework to broader editing tasks.

## 5 CONCLUSION

We propose EraseLoRA, a dataset-free object removal framework that leverages MLLM-guided foreground exclusion and background subtype aggregation. By explicitly preventing non-target foreground reference and softly guiding attention with background cues during inference time, EraseLoRA achieves cleaner and more coherent background restoration across diverse diffusion backbones. Extensive experiments demonstrate its effectiveness and robustness, surpassing prior dataset-free methods both quantitatively and qualitatively.

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
