# OpenReview forum: "EraseLoRA: MLLM-Driven Foreground Exclusion and Background Subtype Aggregation for Dataset-Free Object Removal"
_ICLR.cc/2026/Conference — ICLR 2026 Conference Withdrawn Submission_

### Official Review · Reviewer_ViFF · 2025-10-31

**Soundness:** 3
**Presentation:** 3
**Contribution:** 3
**Rating:** 6
**Confidence:** 3

**Summary:**

he paper proposes EraseLoRA, a dataset-free object-removal framework that plugs into modern text-to-image diffusion models and uses an MLLM to (i) explicitly exclude non-target foregrounds and (ii) aggregate multiple background “subtypes” to fill the hole more coherently.

**Strengths:**

1. The motivation of this submission is clear and the design aligns well with the motavation.
2. The paper is well-written and easy to follow.
3. The qualitative results demonstrated in this submission are impressive.

**Weaknesses:**

1. Evaluation metrics. On OpenImages there is no ground-truth removed image; so they use F-DINO/F-CLIP and B-DINO/B-CLIP. That’s reasonable, but the community may want human or more visual realism metrics. Currently the strongest claims rest on these proxy numbers.
2. Novelty is partly compositional. The proposed method is kind of good engineering on top of existing tools rather than a brand-new learning principle which might not be suitable for ICLR.
3. Robustness experiments on errors from tools such as MLLM mistakes could strengthen this submission.

**Questions:**

My concerns are listed in the weaknesses part.

---

> ### Author Response · Authors · 2025-11-20
>
> We appreciate your recognition of our clear motivation and impressive qualitative results. We would like to briefly share how we are addressing your constructive feedback regarding evaluation metrics, novelty, and robustness in our upcoming revision.
>
> **1. [W1] Evaluation Metrics.** You pointed out that F/B-DINO proxies might be insufficient. To address this, we will expand our evaluation to include paired benchmarks such as RemovalBench. This will allow us to report pixel-level fidelity metrics including SSIM, PSNR, and LPIPS, providing a more concrete assessment of reconstruction quality. Additionally, we plan to introduce a semantic evaluation metric (e.g., GPT-Metric) to assess human-like visual realism and semantic consistency, complementing the current embedding-based metrics.
>
> **2. [W2] Novelty.** We will further clarify that EraseLoRA is not merely an engineering combination but a shift in the object removal paradigm, moving from "attention surgery" to "background-aware reasoning". We will demonstrate through additional experiments that our Foreground Exclusion module (BFE) acts as a fundamental principle that improves even other SOTA methods when plugged in, thereby highlighting the methodological contribution.
>
> **3. [W3] Robustness on MLLM Errors.** To address concerns about dependencies on specific tools, we will conduct extensive robustness tests across various MLLMs with varying parameter sizes (e.g., 7B to 78B). We aim to demonstrate that EraseLoRA maintains consistent performance gains regardless of the MLLM used, verifying that our framework is robust to variations in the reasoning module.

---

### Official Review · Reviewer_JCQs · 2025-10-31

**Soundness:** 2
**Presentation:** 1
**Contribution:** 3
**Rating:** 4
**Confidence:** 3

**Summary:**

The paper proposes EraseLoRA, a dataset-free object removal framework that addresses the issues of unintended non-target foreground regeneration and fine-detail loss caused by disrupted short-range attention in existing dataset-free object removal frameworks.

This framework consists of two stages during Test Time Adaption (TTA):
1. The first stage is Background Reconstruction with Foreground Exclusion (BRF), which distinguishes between foreground and background using an MLLM (Multimodal Large-Language Model) and preserves the background via a background reconstruction loss.
2. The second stage is Background Subtype Aggregation (BSA), which guides the generation of the target region by controlling the cross-attention map of the background through an alignment loss, while avoiding the generation of only a single type of background via a diversity loss.

**Strengths:**

1. The proposed EraseLoRA specifically addresses the issues of unintended non-target foreground regeneration and fine-detail loss caused by disrupted short-range attention in dataset-free object removal through BRF (Background Reconstruction with Foreground Exclusion) and BSA (Background Subtype Aggregation).
2. Compared with directly constraining the attention map, the Test Time Adaption method is more robust.

**Weaknesses:**

1. **Methodological Limitation on Occluded Non-Target Foregrounds.** While suppressing non-target foregrounds effectively avoids their interference during target-region background generation, it also creates a limitation: when non-target foregrounds are occluded by the target foreground, the method cannot reconstruct those occluded parts. This flaw is clearly visible in the provided result figures.
2. **High Computational Overhead from MLLM Dependency.** Invoking an MLLM for each inference significantly increases both computational load and latency, which may limit practical use in resource-constrained scenarios.
3. **Dataset Ambiguity.** Please provide a formal citation for the used dataset. Additionally, clarify if the "ROAD dataset" refers to the existing "RORD dataset".
4. **Inadequate Result Presentation.** The presentation of comparisons with previous SOTA methods is insufficient. Additionally, Figure 7 in the Appendix also lacks the display of input images and masks.

**Questions:**

See weaknesses.

---

> ### Author Response · Authors · 2025-11-20
>
> We are grateful for your constructive comments regarding the methodological boundaries and presentation of our work. We particularly value your acknowledgement that our TTA approach offers a more robust alternative to direct attention manipulation.
>
> **1. [W1] Methodological Limitation on Occlusion.** You raised a valid concern regarding the reconstruction of occluded non-target foregrounds. We will clarify that EraseLoRA handles this specific scenario by leveraging MLLM reasoning. We will demonstrate through new qualitative results that the MLLM identifies occluded objects (e.g., "stairs" hidden behind a person) not as foregrounds to be excluded, but as background tags. Consequently, we will show how the BSA stage guides the model to reconstruct these occluded parts using the inferred semantic cues, ensuring scene continuity.
>
> **2. [W2] Computational Overhead.** We acknowledge that the dependency on MLLM and TTA increases inference cost. We will include a detailed efficiency comparison to show that while our method requires more resources than simple inference, its latency is comparable to other optimization-based dataset-free methods (DesignEdit, AttentiveEraser) while delivering significantly higher background fidelity. We will also discuss potential directions for lightweight latent optimization to mitigate the overhead of TTA.
>
> **3. [W3, W4] Dataset Ambiguity and Presentation.** We apologize for the confusion regarding the dataset naming and result presentation. We will correct the dataset citation to RORD (Real-world Object Removal Dataset). We will expand our evaluation to include recent dataset-driven methods (e.g., OmniEraser, SmartEraser) for a comprehensive comparison. Also, we will ensure that all figures, including those in the appendix, explicitly display input images and masks to facilitate clear visual verification.

---

### Official Review · Reviewer_GR2p · 2025-11-01

**Soundness:** 2
**Presentation:** 2
**Contribution:** 2
**Rating:** 4
**Confidence:** 4

**Summary:**

This paper proposes a dataset free object removal framework using multimodal LLMs to differentiate between foreground from background. It also incorporates a test-time adaptation method for guiding the diffusion. It broadly consists of two stages, namely background reconstruction with foreground exclusion and background subtype aggregation. Experiments on two datasets show improvement over the existing methods.

**Strengths:**

1. Use of MLLMs for foreground-background separation in object removal is novel.

2. Motivation seems convincing.

3. The method is agnostic to the base segmentation models.

4. Strong evaluation protocol.

**Weaknesses:**

1. Gains achieved on the SOTA are only marginal (Tab 2) and at times, loses to dataset-driven methods.

2. Given these marginal gains, statistical significance isn't given.

3. Very large computational cost - Tab 3 shows a significant increase in VRAM.

4. The method seems to be critically dependent on MLLMs. There is no clarity on what happens if and when MLLMs hallucinate objects?

5. Datasets (ROAD) are large enough, and metrics aren't representative enough.

6. There are a lot of missing details in the methodological description.

7. There are no formal theoretical arguments on why losses should impose "puzzle-like" reconstruction.

**Questions:**

1. Report wall-time clock time per image and VRAM for all methods.

2. How does TTA time scale with image resolution?

3. What percentage of test samples have incorrect foreground detection?

4. B-DINO/B-CLIP uses your own MLLM-defined background, which is circular. Provide metrics that don't depend on your methods' intermediate outputs.

5. Statistical significance analysis needs to be done.

6. There is no ablation on the choice of MLLM, Authors should consider doing it.

7. Compare against a simple baseline like masking MLLM detected baselines.

8. Give a detailed failure analysis by showing the cases where MLLM misclassifies, loss doesn't converge, and when there is conflict between background subtypes.

9. Provide details on hyperparamter sensitivity, and number of TTA iterations needed and so on.

10. Given the 2.4× VRAM cost, why not use SmartEraser with lesser params?

11. How does your method perform on cases where multiple objects needs to be removed, where there are unclear boundaries?

---

> ### Author Response · Authors · 2025-11-20
>
> We sincerely thank you for your detailed review and for recognizing the novelty of using MLLMs for foreground-background separation, as well as the strength of our evaluation protocol. We would like to briefly address your concerns regarding performance gains, computational cost, and MLLM dependency in our upcoming revision.
>
> **1. Performance Gains and Comparison.** You raised concerns that gains are marginal and sometimes inferior to dataset-driven methods. We will clearly demonstrate EraseLoRA performance gains and show that EraseLoRA outperforms dataset-driven methods, such as SmartEraser and SDXL-Inpainting, in background fidelity. We emphasize that while SmartEraser has fewer parameters, it requires paired training data; EraseLoRA offers a superior dataset-free alternative for scenarios where training data is scarce.
>
> **2. Computational Cost and Efficiency.** We acknowledge the cost of TTA but wish to clarify the trade-offs. We will include a detailed efficiency table comparing wall-clock time and provide full details on TTA iterations (500 steps) and hyperparameters ($\lambda=0.2$) to ensure reproducibility.
>
> **3. MLLM Robustness and "Circular" Metrics.** We respectfully clarify that our evaluation is not circular. As detailed in our experimental setup, we manually curated ground-truth masks for the test set to ensure reliable evaluation, independent of the MLLM used during inference. To address robustness, we will add a new ablation study across MLLMs ranging from 7B to 78B parameters. In our experiments, EraseLoRA shows consistent gains even with smaller models (e.g., LLaVA-7B), proving the method is not critically dependent on a specific-large MLLM. For hallucination, we will clarify that the BFE stage acts as a filter against hallucinations by cross-referencing MLLM outputs with segmentation models (Tag2Mask).

---

### Note · Authors · 2025-11-20

**Comment:**

We sincerely thank the Area Chair and reviewers (GR2p, JCQs, ViFF) for their time and constructive feedback.

**Withdrawal Confirmation:**

I have read and agree with the venue's withdrawal policy on behalf of myself and my co-authors.